# The Challenging Melanoma Landscape: From Early Drug Discovery to Clinical Approval

**DOI:** 10.3390/cells10113088

**Published:** 2021-11-09

**Authors:** Mariana Matias, Jacinta O. Pinho, Maria João Penetra, Gonçalo Campos, Catarina Pinto Reis, Maria Manuela Gaspar

**Affiliations:** 1Research Institute for Medicines, iMed.ULisboa, Faculty of Pharmacy, Universidade de Lisboa, Av. Prof. Gama Pinto, 1649-003 Lisboa, Portugal; mariana.r.matias@gmail.com (M.M.); pinho.jacinta@campus.ul.pt (J.O.P.); mariapenetra@edu.ulisboa.pt (M.J.P.); 2CICS–UBI–Health Sciences Research Centre, University of Beira Interior, Av. Infante D. Henrique, 6201-506 Covilhã, Portugal; gfrc@ubi.pt

**Keywords:** melanoma, preclinical research, in vitro models, in vivo models, clinical trials

## Abstract

Melanoma is recognized as the most dangerous type of skin cancer, with high mortality and resistance to currently used treatments. To overcome the limitations of the available therapeutic options, the discovery and development of new, more effective, and safer therapies is required. In this review, the different research steps involved in the process of antimelanoma drug evaluation and selection are explored, including information regarding in silico, in vitro, and in vivo experiments, as well as clinical trial phases. Details are given about the most used cell lines and assays to perform both two- and three-dimensional in vitro screening of drug candidates towards melanoma. For in vivo studies, murine models are, undoubtedly, the most widely used for assessing the therapeutic potential of new compounds and to study the underlying mechanisms of action. Here, the main melanoma murine models are described as well as other animal species. A section is dedicated to ongoing clinical studies, demonstrating the wide interest and successful efforts devoted to melanoma therapy, in particular at advanced stages of the disease, and a final section includes some considerations regarding approval for marketing by regulatory agencies. Overall, considerable commitment is being directed to the continuous development of optimized experimental models, important for the understanding of melanoma biology and for the evaluation and validation of novel therapeutic strategies.

## 1. Introduction

Melanoma is an aggressive and often fatal form of skin cancer, and its global incidence is increasing, constituting a serious public health problem [1,2]. This pathology is characterized by the malignant proliferation of melanocytes, cells responsible for melanin production, that can be caused by genetic, epigenetic, and/or environmental factors [1,3,4]. The risk of developing this type of cancer appears to be sex-independent but varies according to the skin phototype and geographic location, with a higher incidence rate in Caucasian individuals and inhabitants of regions with an excessive exposition to ultraviolet (UV) radiation [5,6,7,8]. Moreover, dietary regimens have also attracted attention for reducing melanoma risk. Several antioxidant phytochemicals from vegetables and fruits, for example, have demonstrated chemopreventive properties, whereas alcohol intake can increase the risk of malignancy [9,10]. The role of vitamin D in the management of melanoma should also be highlighted, due to its antiproliferative effects. In this sense, and since solar radiation is essential for vitamin D production, careful sun exposure is recommended [9]. According to worldwide data from the International Agency for Research on Cancer, it was estimated that almost 325 thousand new melanoma cases were diagnosed and more than 57 thousand associated deaths were reported in 2020. Moreover, by 2040, alarming increases in incidence and mortality, 510 and 96 thousand cases, respectively, are expected [11].

In the clinical setting, the most common skin melanoma diagnosis method is based on the ABCDE criteria, by which A is asymmetry, B is the irregular border, C is color variegation, D is a diameter greater than 6 mm, and E is elevated surface of the damaged skin area [12,13,14]. However, since these parameters are not valid for all types of melanoma, other approaches, such as biopsy and imaging techniques, may be used [15,16]. A significant percentage of patients develop metastases that can reach vital organs by spreading from the primary tumor through the lymphatic and circulatory systems [17,18]. Indeed, a century ago, metastatic melanoma was considered a rare form of skin cancer. However, today the average risk of developing this malignancy throughout life has reached 1 in 50 in Western populations [18]. This becomes particularly alarming, since metastatic melanoma is the most aggressive type of skin cancer, accounting for around 75% of skin cancer deaths [19].

Classically, melanoma progression has been represented by the Clark model, a previously widely accepted method for melanoma microstaging, based on the anatomic level of local invasion [20,21,22]. Nowadays, the staging of melanoma considers several parameters, including the Breslow depth, ulceration, extent of regional lymph node invasion, and degree of spreading in the surrounding area and distant parts of the body [23,24,25,26]. As represented in Figure 1, the American Joint Committee on Cancer (AJCC) has defined the TNM system—tumor, lymph nodes, metastasis—for melanoma staging, being internationally recognized and commonly applied [23,26]. Briefly, in the TNM system: (i) T refers to the size of the primary tumor and its spread to adjacent tissues; (ii) N describes the number of regional lymph nodes affected by the tumor; and (iii) M identifies the presence of metastasis. At stage 0, abnormal melanocytes are observed in the epidermis, being designated as melanoma in situ. At stages I and II, localized cancer has formed. At stage III, cancer has spread to nearby lymph nodes, and at stage IV melanoma has spread to other tissues/organs, namely the lungs, liver, brain, spinal cord, bone, soft tissue, gastrointestinal tract, and distant lymph nodes.

Albeit favorable prognoses are attributed to patients diagnosed at early stages of the disease, those in advanced stages entail high mortality and morbidity. When metastasized, melanoma exhibits high resistance to currently available therapies, prompting the search for innovative and more effective therapeutic strategies. In this review, we described the main current therapeutic options for melanoma and summarized the most relevant methods used to screen new drug candidates, focusing on in silico, in vitro, and in vivo studies. In addition, some underway clinical trials were also included, detailing the disease stage and clinical trial phase.

To prepare this review, an extensive literature search was performed using electronic resources. PubMed and Science Direct were the main sources of information, complemented by other information sources, such as Research Gate, and the use of official sources from European Medicines Agency (EMA), and Food and Drug Administration (FDA). The search was carried out between January 2021 and June 2021, aiming to generate a critical and comprehensive overview of the methodologies used for antimelanoma drug discovery and development, as well as for the study of the mechanisms underlying this pathology. Research was occasionally carried out outside these dates. From the articles collected from the initial literature search, an analysis was carried out to select the most relevant ones. The keywords for the search under “Title/Abstract” consisted of combinations of the following terms: melanoma, drug development, computational studies, in vitro, 3D assays, in vivo and animal models. The search regarding clinical trials is described in the respective section.

## 2. Melanoma—Therapeutic Management

The current therapeutic management of melanoma depends on the stage of tumor development, individual characteristics of the patient, and treatment goals. For example, in patients presenting a lower tumor burden or a slowly progressing disease, long-term control of melanoma is required [27].

Surgery is the first option at early stages of melanoma and can be considered curative for melanoma in situ [1,28]. Depending on disease stage, a more extensive surgical procedure may be considered [29,30,31,32]. Another therapeutic modality is radiotherapy, which is often used as a palliative option at advanced stages or when surgery is contraindicated [1]. In terms of chemotherapy, dacarbazine was approved in 1975 by the FDA for the treatment of metastatic melanoma and continues to be considered the reference therapy for this pathology [33,34]. Targeted therapy is based on the specific targeting of genes (e.g., *BRAF*) and/or signaling pathways (MEK) with roles in the process of tumorigenesis, aiming to limit systemic toxicity and improve clinical outcomes [35]. In 2011, vemurafenib was the first of many targeted therapies approved by the FDA [36]. Another therapeutic option is immunotherapy, which is based on the antigen recognition of tumor cells by the innate immune system [37] and can be categorized in biological medications, vaccination, adoptive cell therapy and immune checkpoint inhibitors. Ipilimumab, a human monoclonal IgG1 antibody that targets and blocks cytotoxic T-lymphocyte antigen-4 (CTLA-4), was the first FDA-approved immune checkpoint inhibitor for metastatic melanoma. Further, nivolumab and pembrolizumab, which target the interaction between programmed cell death 1 (PD-1) and its ligands PDL-1 and PDL-2 [38,39,40], were also approved for clinical use. The main advantages and limitations of each therapeutic option are described in Table 1.

## 3. Drug Discovery and Development (Preclinical Research)

Although a panoply of therapeutic strategies is available for melanoma, an ideal drug has not yet been found. While early-stage melanoma can often be treated by surgical excision, metastatic melanoma, which is associated with high aggressiveness and low survival rates, remains untreated in most cases. In this sense, new therapies are needed. In general, prior to becoming available for commercialization, new drugs must pass through several steps, namely discovery and development, preclinical studies, clinical trials and review by regulatory entities [41,42,43]. Research and discovery of new antimelanoma molecules may start by preliminary selection of the most promising lead compounds through computational approaches (in silico) [44,45]. This strategy is usually followed by in vitro testing of hit compounds, selection of the most promising for further in vivo studies and, ultimately, clinical trials, as described in Figure 2 [46]. Taking into consideration that a drug discovery and development program lasts 1–2 decades and costs around $3 billion, it is disappointing that around 60% of new anti-cancer agents fail in advanced randomized controlled trials. In addition, the success rate from first-in-human to registration in oncology field is approximately 5%, being much lower when compared to other pathologies [47]. For this reason, the establishment of more accurate and reliable methodologies to be used at early drug discovery and preclinical research represents an important strategy for a successful selection of potential compounds. This ultimately reduces the associated costs and attrition rates that occur at clinical development phases. In the following sections, the in silico, in vitro, and in vivo assays used for the discovery/development of new drugs towards melanoma will be addressed.

### 3.1. In Silico Models

The use of computational approaches for anticancer drug discovery and development has become increasingly popular in the last few decades [48]. Computational tools require the implementation of an interdisciplinary approach, where mathematical and computational methods provide new and useful information [49,50]. For instance, the FDA-approved binimetinib and encorafenib antimelanoma drugs benefited from in silico methods during their development [51]. Computational tools enable a high understanding of melanoma pathogenesis and the prediction of the potential anticancer activity of lead compounds [48,50]. Data obtained from virtual analysis may be useful in the investigation of the complex signaling networks involved in multifactorial diseases, such as melanoma. Briefly, they can be used for different purposes, such as the identification of novel targets, elucidation of intracellular signaling pathways, simulation of organ/tissue level behavior, and analysis of therapy resistance mechanisms [50]. Furthermore, computer-aided drug discovery allows the screening and selection of promising lead candidates from more than several millions of compounds, considerably reducing the number of more expensive and time-consuming experimental assays performed. Additionally, in silico predictive models also proved to be efficient in the elucidation of structure–activity relationship problems and prediction of the pharmacokinetic and pharmacodynamic profiles of tested compounds [48,50]. These computational approaches are commonly categorized in structure- and ligand-based methods, which will be described in the following sections. It is worthy to note that both methods have been used in an integrative way, potentiating the strengths and reducing the limitations of each one [48,52,53,54].

#### 3.1.1. Structure-Based Approaches

Structure-based methodologies rely on the known structural layout to ascertain the interaction between compounds and the target [51]. This provides a starting point to simulate interactions between compounds from virtual libraries with macromolecular receptors, further enhancing its affinity [52]. The understanding of binding site interactions at the therapeutic target has been drastically improved through the use of biomolecular spectroscopic technologies, such as nuclear magnetic resonance and X-ray crystallography. With the increase of readily available and reliable 3D structures of macromolecules, the discovery of new anticancer drugs was accelerated by the design of structurally diverse new ligands [48,51].

Molecular docking is a typical structure-based strategy for rational drug design, allowing the prediction and study of the most efficient ligand conformation and orientation at the binding site. With the advancement of computational resources, docking methods have become more easily accessible [51]. Using this approach, Couto and collaborators found novel candidates against metastatic melanoma, particularly tetra-cationic platinum(II) porphyrins. Obtained data suggested their affinity for the N-terminal region of Apolipoprotein B-100 of the low-density lipoprotein receptor [55]. Another study demonstrated the potential of valproic acid as a potential anticancer agent towards B16-F10 resistant melanoma cells. Molecular docking studies predicted two stable interactions between the drug and the Arg39 of histone deacetylase 2, the overexpression of which has been associated with several types of cancer, including melanoma [56,57]. In addition, vanicosides A and B demonstrated interesting anticancer activity towards C32 melanoma cells, possibly due to binding at the active site of BRAF V600E and MEK-1 kinases [58]. Additionally, new pyrimidine–pyrrole derivatives with substituted 1,2,3-triazole nucleus, displaying IC_50_ values around 13 µM against B16-F10 melanoma cells, were found to bind EGFR tyrosine kinase (through the interaction with Cys797 and other residues), as suggested by in silico docking assays [59]. Moreover, cinnamic acid derivatives and Triangularin showed in vitro antimelanogenic activity, probably due to interaction with the catalytic site of mushroom tyrosinase, through hydrogen bonds with Arg 268 and His 85, respectively [60,61].

Another strategy is the structure-based pharmacophore, which relies on the analysis of the complementary chemical features of an active site and its spatial restrains and relationships. It can be classified into target–ligand-complex-based and target-binding-site-based methodologies. The first approach can conveniently simulate the position of the ligand at the target (e.g., protein), assessing key interactions between them. On the other hand, the target-binding-site-based method is ligand independent and interactions with the binding site can be defined according to pharmacological characteristics [51,52]. An example of the structure-based pharmacophore approach was presented by Jha and collaborators, who developed a virtual screening protocol that, combining molecular docking and molecular dynamics simulations, led to the identification of monoacylglycerol lipase inhibitors. This enzyme has been associated with melanoma invasion and progression [62,63].

#### 3.1.2. Ligand-Based Approaches

Ligand-based drug design has been mainly employed when little or no reliable structural information on the drug targets is present [48]. Therefore, this approach is based on the concept of molecular similarity, by which molecules with comparable structures tend to cause equivalent biological effects. Opposed to structure-based approach, the template is the compound itself [64]. These methods are commonly used to screen new ligands with putative biological activities or for pharmacokinetic profile optimization. Indeed, the most widely used technique relies on calculated molecular descriptors, such as those related to physicochemical properties (e.g., molecular weight, octanol-water partition coefficient or surface areas), two-dimensional (2D) fingerprint and three-dimensional (3D) shape similarity searches [51]. More complex techniques, such as pharmacophore modeling with known ligands and quantitative structure activity relationship (QSAR) models, are predictive models that can be suitable for drug discovery and optimization [52,65]. In the first case, structural overlap of key molecular features from active compounds or binding site pocket representations are used as a spatial layout to represent the most probable location of chemical characteristics and additional geometrical constraints [52]. Regarding the QSAR method, it is a popular ligand-based model consisting of an analysis of biological activities through a set of molecular descriptors or fingerprints, correlating the biological activity experimentally measured with the properties of the ligand. QSAR has been applied for the prediction of biological activity and discovery and optimization of lead compounds [51,52,66].

Although limited, some examples make use of ligand-based approaches at early stages of drug discovery [67,68,69,70]. A dataset of seventy-two molecules was analyzed by QSAR techniques to investigate their activity against the human melanoma cell line SK-MEL-2. The generated model presented high activity–descriptor relationship efficiency (correlation coefficient of determination (R^2^) = 86.4%) and a good activity prediction efficiency, according to the cross-validated regression coefficient Q^2^_CV_ (79.9%) [53]. In addition, Anbar et al. described potential antimelanoma compounds, incorporating an imidazothiazole nucleus with selective activity against V600E mutant BRAF kinase. 3D QSAR studies were included in the investigational work, aiming to understand the contribution of structural features for the observed activity. A three-latent variables model (R^2^ = 0.857), comprising three interaction fields encoding shape, hydrophobic and hydrogen bond acceptor regions was considered to be the best model [68]. The 3D-pharmacophore and 2D-QSAR modeling techniques were also used to study the antimelanoma activity of a set of spiro-alkaloids derivatives [69].

### 3.2. In Vitro Models

In vitro assays (often cell-based assays) are carried out outside living organisms, under controlled conditions, and are usually used to evaluate the cytotoxic potential of compounds, as well as their underlying mechanisms of action [46,71]. In the current preclinical pipeline of anticancer discovery, high-throughput screening (HTS) in vitro assays, together with combinatorial chemistry, have been considered the prototype strategy to rapidly identify agents with therapeutic potential [72,73]. Nowadays, a wide range of in vitro assays are available, as depicted in Figure 3. In the next sections, these assays will be described, being divided in two groups: 2D and 3D in vitro tests.

#### 3.2.1. 2D Models

Most studies of melanoma cell biology and preliminary screening to identify potential compounds have started from 2D adherent cell culture assays. Usually, they grow as monolayers on tissue culture plates, with relatively high levels of oxygen and nutrients [74]. There are more than 2000 melanoma cell lines established by many laboratories, which makes this pathology one of the most studied in the cancer field [75,76,77]. Within these, around 200 human melanoma cells are perfectly characterized in terms of genetic aberrations, gene expression patterns, and biological properties [77]. Indeed, both melanotic (e.g., COLO829 and TXM-13) and amelanotic (e.g., A375, C32 and SK-MEL-28) human melanoma cell lines are commonly used in in vitro screening assays. They are used to select drug candidates, as well as to understand their efficacy towards melanoma [78,79,80,81]. In addition, patient-derived cells, directly obtained from biopsies of both primary and metastatic tumors, have been included in preclinical research, since they most closely represent the tumor heterogeneity and melanoma aggressiveness [46,82,83].

Beyond humans, several murine cell lines have also been established to be used in immunocompetent mice. One of the most studied is the B16-F10 cell line, which is derived from melanoma induced in C57BL/6 mice [84,85,86]. These cells are highly metastatic and present strong pigmentation [87]. Other examples of murine melanoma cells are K1735-M2 and YUMM [88,89]. Although useful, murine cells display some differences from their human melanoma counterparts, not entirely reflecting human disease. For instance, dissimilarities are found regarding adhesion and growth factor profiles, invasion processes, and antiapoptotic mechanisms [77]. Despite these differences, it is noteworthy that an “ideal model” does not exist. In this sense, both human and murine melanoma cells are extensively investigated, aiming to achieve the most reliable results [77].

Indeed, both human and murine cells are routinely used in 2D assays and, frequently, compounds that do not show effects in 2D cultures might not be effective in more sophisticated in vitro or in vivo models, thus rendering a suitable option for initial screening of drug candidates. In these 2D culture assays, cytotoxicity, migration, invasion, protein expression, molecular characterization, and genetic/genomic characterization are routinely investigated [75,90]. Within cytotoxicity tests, the most popular is, undoubtedly, the (3-(4,5-dimethylthiazol-2-yl)-2,5-diphenyltetrazolium bromide) (MTT) assay towards B16-F10, MNT-1 and A375 cell lines [91,92,93,94]. Notwithstanding, other colorimetric methods have also been used to evaluate cytotoxicity, such as the (3-(4,5-dimethylthiazol-2-yl)-5-(3-carboxymethoxyphenyl)-2-(4-sulfophenyl)-2*H*-tetrazolium) (MTS) assay [95,96,97,98,99]. Other relevant 2D models are usually employed to assess the mechanisms of action of selected candidates. An example was explored by Chen and coworkers, who evaluated the activity of glaucocalyxin A in A375 and A2058 human melanoma cells. They reported that this natural compound inhibited cell proliferation, arresting cells in the G2/M phase of the cell cycle, induced the overproduction of cellular reactive oxygen species and decreased the mitochondrial membrane potential, as demonstrated by flow cytometry assays. Further, Western blot showed an upregulation of the Bax/Bcl-2 ratio and a decrease of NF-κB/p65 phosphorylation [100]. Another natural product, shikonin, also displayed anticancer in vitro activity towards the A375SM melanoma cell line. Both DAPI staining and flow cytometry showed that shikonin could exert its cytotoxic effect through apoptotic pathways. This was also confirmed by Western blot analysis, which revealed high expression of Bax and low expression of Bcl-2 and the involvement of the mitogen activated protein kinase (MAPK) pathway [101]. S-petasin also displayed interesting activity in the MTT assay using A375 and B16-F10 cell lines, as well as inhibition of the migration and invasion, as proved by wound healing and transwell cell assays, respectively. Western blot analysis demonstrated that s-petasin activated the p53 pathway signaling and regulated the expression of Bcl-2, Bcl-XL, Bax, matrix metalloproteinase (MMP)-2, MMP-9, p21, CDK4 and cyclin D1 [102].

Considering that melanogenesis is an important process associated with melanoma, this pathway has attracted attention [103]. In this context, tyrosinase is the key enzyme involved in the first step of melanin synthesis and its overexpression is widely recognized in melanoma [3,104]. Taking this in mind, Choi and collaborators demonstrated that compounds containing the β-phenyl-α,β-unsaturated carbonyl scaffold exhibited potential antimelanoma properties, acting through the inhibition of tyrosinase activity and suppression of melanin production in B16-F10 melanoma cells [105]. Moreover, a set of triazene prodrugs has also shown antiproliferative properties towards MNT-1 and SKMEL-30 human melanoma cell lines expressing tyrosinase [106].

An important advantage of the 2D models (Table 2) is the fact that the microenvironment is free from non-melanoma cells that, by interfering with RNA expression or protein synthesis, could change the analysis and understanding of melanoma-specific pathways [75,90]. Nevertheless, co-culture assays are also important when the aim is to study the biology of melanocytes within a microenvironment that most closely simulates the in vivo setting. In this context, when melanocytes are co-cultured with keratinocytes, they exhibit a phenotype similar to the one observed in vivo, compared to melanocytes in monoculture [90]. Overall, traditional 2D platforms provide a valuable contribution to in vitro cell biology study and have led to several landmark discoveries, including the predominance of BRAF V600E mutations in melanoma [107]. However, these models cannot accurately mimic the complex 3D architecture of the extracellular matrix, where the native cells reside in vivo [74] (Table 2).

#### 3.2.2. 3D Models

In the last decade, the scientific community witnessed a growing improvement in cell culture conditions and an accelerating implementation of 3D models at early drug discovery. The growth of cancer cells in 3D matrices has led to the development of multidimensional structures, which more closely resemble in vivo tumors from which they are derived. In addition, they provide more information about new chemical entities in advanced stages of drug development, as well as accurate predictions of patient’s responses to therapy [108,110]. Thereby, 3D cultures have overcome the lack of a microenvironment found in 2D cultures, while mimicking, to a certain extent, the high complexity of in vivo melanoma models, presenting both ethical and economic advantages [74,90]. There are several 3D models that have enabled studying of melanoma biology, which can be used to select more efficacious and safer antimelanoma therapies. Among these type of models, the following are highlighted: (i) spheroids produced either with cancer cells in monoculture or by combining cancer and stromal cells; (ii) tumorospheres generated with cancer stem cells; (iii) organoids composed by multiple cell types, preserving tissue architecture and diversity; (iv) skin reconstruction, melanoma-on-chip and bioprinting, reproducing a dynamic reconstructed skin and mimicking tumor cell architecture; (v) neoangiogenesis addressing tumor vasculature [74,111].

All of these models will be described in detail in the next sections, and the respective advantages and limitations are indicated in Table 2. Different principles and protocols may be applied to produce these sophisticated cell cultures, mimicking to a greater extent the tumor heterogeneity observed in vivo and leading to preclinical outcomes that are reliable and easy to interpret [110].

##### Spheroid Model

Multicellular tumor spheroids were initially developed by Sutherland and collaborators in 1970 [118] and, currently, are probably the most popular 3D model used for drug screening. Spheroids are aggregates of cells that may reach a mean size between 0.5–1 mm^3^ [111]. This model aims to recreate the behavior of melanoma in vivo, mimicking the oxygen/nutrient gradient, lactate accumulation, and ATP distribution [90,111,119]. A spheroid is divided into three areas: the necrotic/apoptotic core, followed by a quiescent cell layer and, finally, an external area where proliferative cells are located [111]. Different cells, displaying distinct growth and invasion characteristics (e.g., melanoma cell lines, human-primary- and metastasis-derived cells), can be used to generate spheroids [90,120,121]. Melanoma cells from tumors at early stages form spheroids that generally do not invade the matrix, displaying a radial growth phase phenotype. In contrast, cells from metastatic melanomas are highly invasive, showing a vertical growth phase phenotype [75]. Moreover, immune cells, stromal cells, keratinocytes, fibroblasts, or endothelial cells are often used in tumor co-cultures [74,122,123,124]. However, in order to adequately simulate the original tumor, it is important to carefully consider the adequate number of cell types to be included in a 3D system [111].

Throughout the years, multiple approaches have been proposed for spheroid generation. Generically, they can be classified as scaffold-free and scaffold-based systems. In the first case, melanoma cells spontaneously aggregate to form spherical 3D structures when they are placed in an environment where cell–substrate interactions are dominated by cell–cell interactions. The scaffold-based strategy involves the use of a porous 3D scaffold, which physically supports cell aggregation, allowing the formation of multicellular spheroids with a controlled size [74]. Additionally, different matrices can be used to recreate the natural tumor microenvironment. Collagen (mainly type I) is the most commonly used, since it is the major constituent of extracellular matrix in most organs, and it is easy to manipulate in terms of elasticity and stiffness [125,126,127,128,129]. Other options are, for example, matrigel, fibrin, or agarose [129,130,131].

To evaluate the efficacy of new compounds or study melanoma pathophysiology, several tests can be carried out on 3D spheroid models, such as MTT/MTS assay, measurement of spheroids area, apoptotic and live/dead viability assays, as well as the study of tumor metabolism, progression, invasion, and drug resistance [74,132,133,134,135]. Various reports have addressed the differences between results obtained using 2D and 3D spheroids models. For example, Hundsberger and collaborators evaluated the effect of quercetin in both models. Interestingly, this natural product reduced cell viability in 2D melanoma cells (1205 Lu cells, MCM 1G, and MCM DLN), whereas the same cell lines in the spheroid form were much less sensitive to the tested compound. These findings are important, since 3D platforms display proliferation rates and morphologic features that most closely resemble the complexity and growth behavior of human melanoma [136]. Similar results were found to the compound bis-anthracycline WP760, which was tested against a panel of melanoma cell lines originating from tumors at different developmental phases. In 2D cell cultures, WP760 displayed more potent inhibition of cell proliferation compared to spheroid models [137]. In another research work, a set of antitumor peptides derived from lactoferricin demonstrated cytotoxic effects towards A375 cells, both in 2D and spheroid forms. The most active peptide in the 2D model presented just marginal effects in 3D cultures, probably due to its high hydrophobicity that prevented the peptide from reaching the spheroid core [138]. On the opposite, a lipophilic bisphosphonate inhibited spheroid growth in 3D cultures of A375, A2058 and VM47 melanoma cells, being in accordance with data obtained in 2D models (IC_50_ values of 4.4, 5.9 and 2.9 µM, respectively) [139]. Another recent work reported the synergistic effect of afatinib (an ERBB inhibitor) and crizotinib (a MET inhibitor) in 3D spheroid models using 13 different cutaneous malignant melanoma cells. The cytotoxic effects were similar to those found in the respective 2D cultures. This co-treatment also induced apoptosis, based on increased levels of cleaved caspase-3, as confirmed by Western blot assay in the spheroid model [140].

##### Tumorosphere Model

Although multicellular spheroids replicate the tumor microenvironment, they do not allow to isolate and expand certain tumor cell subpopulations, such as cancer stem cells, which have been associated to tumor initiation, progression, and recurrence [74,141]. To study these types of cells, the first in vitro model was reported by Singh and co-workers, who expanded neural stem cells in a 3D culture system [142]. In general, the tumorosphere formation assay consists of culturing cells at low density in specific media (e.g., stem cell medium, serum-free and supplemented with basic fibroblast and epidermal growth factors) under low-adherent conditions [115,143,144]. The resulting spherical aggregates are derived from the clonal expansion of one single cell, instead of cell proliferation [115,143]. The population can be enriched with several cancer stem cells markers, such as CD44 [141,145,146], CD24 [141], angiopoietin-like 4 [147], CD20, CD133, and Wnt-3A [145].

Tumorospheres are used to investigate the self-renewal ability of cancer stem cells and provide an adequate platform for screening potential anticancer agents [114]. For example, Marzagalli et al. reported that δ-tocotrienol, a Vitamin E derivative, markedly reduced the formation of new melanospheres. Although this compound did not induce the complete disaggregation of pre-formed melanospheres, lower mean size, border irregularity and lack of compactness were observed, when compared to control spheres [148]. In addition, morin, a natural ingredient isolated from the Moraceae family of plants, significantly reduced the sphere formation activity of CD133^+^ melanoma MV3 cell subpopulation. This study also showed that cells overexpressing miR-216a demonstrated less sphere formation activity, compared to the negative control [145]. Mukherjee and collaborators investigated whether the combination of a γ-secretase inhibitor and a small molecule Bcl-2/Bcl-xl/Bcl-W inhibitor could be a good strategy to overcome the pharmacoresistance phenomenon. Indeed, they found that the combination treatment was efficient on disrupting primary spheres and almost eliminating all secondary sphere formation of tested melanoma cell lines at a higher extent than the single agents [149].

##### Organoid Model

Organoids represent a more complex ex vivo 3D model able to self-propagate in an extracellular matrix, including autologous lymphoid, myeloid, and other host cell populations [75]. They arise from stem or slightly differentiated cells that are obtained from primary tissues collected from surgical resection specimens or core needle biopsies [111,150]. To obtain the organoids, collected tissues are often subjected to classic digestion methods (mechanical or enzymatic). However, this process may disrupt native 3D growth architectures, with loss of tumor infiltrating immune cells [150]. Nevertheless, peripheral blood mononuclear cells or other immune cell subsets can be, subsequently, added as a co-culture [111]. In this context, Votanopoulos et al. developed a 3D mixed tumor/node organoids and tumor/peripheral T-cell organoids from tissue biospecimens sets obtained from patients with melanoma at stages III or IV [151]. Moreover, a co-culture of organoids and autologous lymphocytes was also carried out through isolating peripheral blood mononuclear cells from patients with advanced melanoma. The authors reported that the tumor–T-cell interaction induced FKBP51s, an immunophilin highly expressed by immune cells, which is important for the selection of candidate patients to melanoma immunotherapy [152]. Although advantageous, organoids also have some limitations, such as the lack of one or multiple compartments (e.g., the vascular system) (Table 2) [111].

##### Skin Reconstruct Model and Bioprinting

Skin reconstruct model consists of artificial skin composed of a stratified and differentiated epidermal compartment of keratinocytes and melanocytes/melanoma cells and a dermal compartment containing fibroblasts embedded in collagen. The cells can be derived from human skin or embryonic/induced pluripotent stem cells. This skin model mimics the complex organization of the in vivo tissues, closely resembling human skin in architecture and composition, including the major cell types at physiological ratios [75,90,153,154]. Skin reconstruct simulates stage-specific properties of melanoma cells, exhibiting the same characteristics observed in human cancer skin: the growth of cells derived from a melanoma in situ is limited to the basement membrane, whereas advanced primary and metastatic melanoma cells grow through the basement membrane, invading the dermis [90,155,156]. Moreover, to improve therapeutic outcomes in elderly melanoma patients, a 3D skin reconstruct model was established to reflect the characteristics and physical changes of aging skin [157].

The skin reconstruct 3D model considers the contributions of the stroma and surrounding cells to melanocyte proliferation and differentiation. For this reason, it has been used to study cell–cell interactions and the effects of the environment in melanogenesis regulation, as well as in the proliferation and differentiation of keratinocytes [74,158]. For example, Yang and coworkers demonstrated that a combination of the MITF, SOX10 and PAX3 factors enabled the conversion of murine and human fibroblasts to functional melanocytes, which may be useful for melanoma etiology investigation [159]. The skin reconstruct model has also been employed to assess the progression and invasion of tumors, as well as to evaluate the antimigratory effects of different drug candidates, reducing the time and costs associated to the animal experiments [74,153]. An example is given by Michielon et al., who developed a human melanoma skin reconstructed model to study tumor progression and invasion. This 3D model simulated the early stages of melanoma invasion, demonstrating the local progression of SK-MEL-28 melanoma cells. The authors also reported MMP-9 expression consistent with early invasive events and an increase of CXCL10 and IL-10 secretion, indicating an immune suppressive effect [160]. Regarding drug assessment, the compound 22β-hydroxytingenone decreased melanoma invasion in a SK-MEL-28 skin reconstructed model in a higher extent, when compared to control. This reduction was attributed, at least in part, to the reduction of MMP-9 activity [161]. Using the skin model, the compound 2-methoxyestradiol reduced invasion and migration of melanoma cells, since they were only scarcely detected in the dermis compartment [162]. Moreover, the broad spectrum MMP inhibitor Ilomastat led to a marked decrease in cell invasion in a human skin reconstruct model of SK-MEL-28 melanoma cells resistant to vemurafenib [128]. An additional study, using A375 melanoma cells, showed that a combination of BRAF, MEK, and aurora kinase A inhibitors impaired the formation of tumor nodules close to the epidermis and reduced the invasion of dermal structures [163]. A neobavaisoflavone, isolated from the plant *Pueraria lobata*, markedly reduced melanin contents in a reconstructed human 3D skin model at the same or higher extents than arbutin, which is a melanin synthesis inhibitor [164]. Similar results were found to pyruvate, which demonstrated antimelanogenic activity in a human pigmented epidermis skin model [165].

Another 3D model uses additive manufacturing and is designated as bioprinting. This consists of printing live cells, extracellular matrix components and biocompatible materials in complex 3D living tissues to generate the desired organoid architecture, topology, and functionality [74,110]. Important steps in this model include the selection of the right printer and tissue structure design, the choice of a printable material, the definition of cell types and densities, and the integration of all these aspects in a viable and functional tissue [166,167]. For example, Schmidt et al. demonstrated that melanoma cells present different behaviors according to the selected matrix [168]. Bioprinting allows the creation of realistic and geometrically complex morphologies and can be used in HTS, demonstrating reproducibility, high precision and accuracy, co-culture ability, low probabilities of contamination and a reduced specialized training [74,167]. On the other hand, the lack of vascular and lymphoid systems is the main limitation of this model [74,169].

##### Melanoma-on-Chip Model

The organ-on-chip models replicate, in vitro, the complex microenvironment, architecture, and function of the human organs, overcoming some of the organotypic culture limitations, such as inefficient tissue perfusion [74,110]. An organ-on-chip is a microfluidic device where a variety of living cells can be cultured and continuously infused in micrometer-sized chambers, allowing for controlled release of growth factors, nutrients, or even drug candidates. The integration of microfluidics and electrical sensing modality in 3D tumor microenvironments may enable accurate and rapid monitoring of cancer cell responses to series of drugs. In addition, the implementation of nanotechnology-based microfluidics has given the possibility of exploring cell interactions at a microscale level in physiologically dynamic environments [74]. Thus, this model allows us to explore several parameters of carcinogenesis, and is also useful for drug efficacy screening [111]. In this context, Ayuso and collaborators developed a microfluidic device with air walls to control the patterning of WM-115 melanoma cells, when in co-culture with dermal fibroblasts and keratinocytes. They showed that the presence of fibroblasts and keratinocytes promoted alterations in melanoma cell morphology and growth pattern, as well as in chemokine secretion. The important role of the skin microenvironment on melanoma progression was proved by the metabolic shift in melanoma cells due to the presence of stromal cells [170]. Researchers have also developed a microfluidic co-culture device containing two compartments separated by a hydrogel barrier. This allowed a better understanding of signaling pathways in the melanoma microenvironment and supports the development of more efficacious anticancer therapies. For example, this model was established using melanoma cells that were sensitive (A375 and LOX-IMVI) or resistant (LOX-IMVI-R) to vemurafenib [171].

##### Neoangiogenesis Model

Neoangiogenesis is the process by which tumor cells promote the formation of new vasculature through the secretion of pro-angiogenic factors, providing the required nutrients and oxygen and eliminating metabolic substrates. Thus, this newly created vasculature has been considered essential for tumors growth and progression, including melanoma [90,172,173]. As opposed to normal blood vessels, tumor vasculature is disorganized and leaky, being recognized as an important therapeutic target [112,172]. To address the antiangiogenic activity of drug candidates, a simple 3D model was developed through the co-culture of tumor cells and stromal fibroblasts, which interact to form the vascular network in a 3D collagen matrix. It has been described that each melanoma cell line releases distinct growth factors and matrix proteins, leading to the generation of models with different behaviors [75,112]. Moreover, it has been suggested that normal skin-derived fibroblasts influenced melanoma-supported angiogenesis in a collagen matrix to a higher extent than what observed in human melanoma intrinsic biology [174]. The neoangiogenesis models enable the research of pro-angiogenic mechanisms, thus allowing the design of new therapeutic options. For example, tivantinib disrupted the vasculogenic mimicry (VM) exhibited by human melanoma cells (C8161 and WM793) in a 3D matrigel matrix [175]. VM has been associated with highly aggressive primary and metastatic melanoma, being characterized by the formation of new blood-vessel-like structures [176,177,178]. They are formed by cancer cells resting on an inner glycoprotein rich matrix (without endothelium), whereas blood vessels are constituted by a monolayer of endothelial cells surrounded by a basement membrane [176]. In this sense, apatinib also led to the formation of a lower number of VM structures, compared to control, in a 3D model containing human MUM-2B melanoma cells [179]. In addition, nicotinamide inhibited the formation of VM vessel-like structures and destroyed those already formed in a 3D model using primary melanoma cells grown in a matrigel matrix [180].

Undoubtedly, in vitro models provide crucial data about drug biological activity, target interaction and mechanistic action. However, despite recent advances, these studies are not suitable for a perfect reproduction of melanoma complexity, organization, and microenvironment in vivo. For this reason, animal models are a key factor of the drug development program.

### 3.3. In Vivo Models

Appropriate melanoma animal models are extensively used to assess the biological relevance of therapeutic targets, to evaluate tumor response to therapy, to determine pharmacodynamic and pharmacokinetic profiles, to define the maximum tolerated and first-in-human doses, as well as to identify potential disease progression biomarkers [73,181,182]. The use of animals in preclinical research provides similar complex biological interactions and physiological features to those observed in humans, which is of major importance when studying a multifactorial disease such as cancer [73,183]. Depending on the research purpose, different animal models are available. In the next subsections, we will highlight mammalian models for melanoma, including murine and dogs, as well as other relevant models, namely zebrafish.

#### 3.3.1. Murine Models of Melanoma

Mice models are the most used in biomedical research because they are mammals, small sized and easy to manipulate and breed [181,184]. Here, as depicted in Figure 4, the following murine melanoma models will be addressed: syngeneic (Figure 4a), xenograft (Figure 4b), genetically engineered, and UV/carcinogen-induced (Figure 4c).

##### Syngeneic Model

Syngeneic are models (Figure 4a) where murine cells are inoculated into animals with the same genetic background. The B16 syngeneic mouse model is, undoubtedly, the most widely used, because of its well-known genetics and similar histological characteristics to human melanoma [46,90]. This cell line, derived from C57BL/6 mice, has given rise to a panoply of subclones with different predisposition relating to proliferation, invasion, and metastasis. While B16-F1 is not prone to metastasize, B16-F10 is characterized by high metastatic potential to distant visceral organs, in particular the lungs [185].

The syngeneic model allows the study of tumor growth and progression in immunocompetent mice, taking into account the interactions between melanoma and immune cells [109,186]. Moreover, this animal model has been used for the screening of a large number of drug candidates, as well as the study of melanoma pathophysiology and metastasis [46,185]. Cell inoculation can be performed subcutaneously (s.c.), intraperitoneally (i.p.) or intravenously (i.v.), depending on the purpose of the study. The s.c. injection of melanoma cells leads to the formation of a primary tumor that can be easily monitored by macroscopic visualization. In a study conducted by Haratani et al., the authors inoculated s.c. B16-F10 cells in C57BL/6 mice. They demonstrated that the oral administration of nintedanib markedly delayed tumor growth (around three-fold) and prolonged mice survival when compared to control [187]. Using the same tumor induction protocol, Ferreira and coworkers demonstrated that indomethacin incorporated in mesoporous silica nanoparticles inhibited tumor growth by up to 70%. This effect could be explained, at least in part, by the increased levels of cleaved caspase-3 (156%), compared to control (around 50%) [188]. In another study also using a B16-F10 syngeneic murine model, gemcitabine incorporated in lipid-coated calcium phosphate nanoparticles reduced the expression of survivin and Bcl-xL in tumors, compared with the free drug and control groups [189]. Furthermore, the therapeutic potential of liposomes encapsulating the copper complex, Cuphen, was evaluated using a similar murine model. Tumor progression was significantly impaired by treatment with Cuphen pH-sensitive liposomes, compared with control and with mice receiving the free complex [95]. The B16-F1 cell line, displaying a lower metastatic potential, has also been injected s.c. into C57BL/6 mice to induce tumor formation. Following this protocol, co-treatment with S-adenosylmethionine and an anti-PD-1 significantly reduced tumor volume and tumor weight, 315 mm^3^ and 0.2 g, respectively, in comparison with control, 1020 mm^3^ and 0.68 g, respectively [190]. Moreover, in a B16-F1 C57BL/6 model, compound C, a reversible inhibitor of AMP-activated protein kinase, also reduced tumor growth (around 2.5-fold, compared to control) and angiogenesis [191].

The use of models that mimic invasiveness are also important in melanoma drug discovery, since this malignancy is characterized by its high aggressiveness and ability to metastasize. In this context, i.v. injection of B16 cells to obtain pulmonary metastases has been widely used as a model to investigate the effect of new molecules on preventing or reducing metastases formation [46,192]. An example is given by Liu et al., who demonstrated that tegaserod, a serotonin agonist, significantly reduced the number of lung metastases, compared to control, in B16-F10 C57BL/6J mice model [193]. Additionally, in a B16 syngeneic mouse model, the combination of phyto-sesquiterpene lactone deoxyelephantopin and cisplatin inhibited the formation of pulmonary melanoma metastases, without the severe renal side effects observed for cisplatin alone [194]. Isoxanthohumol also inhibited the lung metastases formation in C57BL/6 mice inoculated i.v. with B16-F10 cells [195]. One important advantage of this model is the extremely rapid formation of lung metastases. However, the rapid onset of the disease does not mimic the actual events observed in melanoma patients, where metastasis originate from primary tumors. When cells are intravenously injected, the first steps of the metastization process, namely localized invasion and intravasation into the blood vessels, are bypassed (Table 3) [46].

In addition to the models of lung metastases, murine melanoma cells can also be injected directly into the spleen, leading to the formation of hepatic metastases. This protocol was carried out by Seitz and collaborators, who inoculated B16-F10 cells into the spleen of syngeneic C57BL/6N mice and found that xanthohumol, a natural flavonoid, reduced hepatic tumor burden, as well as the number of hepatic metastases [196]. Similar results were found for compound JR-AB2-011, a mTORC2 specific inhibitor, using the same tumor induction protocol [197].

Other cell lines have been employed in syngeneic transplantation models. For instance, melanin producing Harding–Passey cells are obtained from the dermal melanoma of ICR mice and BALB/c × DBA/2F1 mice, being used in studies involving the effects of melanin content on the metabolic function of melanoma [185]. In addition, the murine B78-D14 melanoma cell line has also been used for immunotherapy studies, either through intradermal or s.c. inoculation. These cells are derived from B16-F10, but the tumor growth is slower and no metastases are formed [198]. Murine B16-OVA [199,200], SM1 [201] and *BRAF* mutant D4M3.A and YUMM1.7 [202] melanoma cell lines are other examples that have been used to test in vivo the therapeutic effect of drug candidates.
cells-10-03088-t003_Table 3Table 3Main melanoma murine models: advantages and disadvantages.Experimental ModelAdvantagesDisadvantagesReferencesSyngeneicFunctional immune system.Fast and easy to establishTumor interaction with the microenvironmentMetastasis formationBoth tumor cells and mouse with the same genetic backgroundLess predictive for clinical translationDifferent anatomy, physiology and biochemistry compared to human (e.g., adhesion proteins and growth factors)Not properly reproducing the interactions between cancer cells and the immune systemLimited availability of cell linesRapid and uncontrolled cell growth[73,90,109,186]XenograftUse of human tumor samplesHeterogeneityMetastasis formationSimple to accomplishPossibilities for “co-clinical trials”Study of drug resistanceLarge number of available human cell linesTumors are easily and precisely measuredTime-consumingExpensive (compared with immunocompetent mice)Lack of immune systemPoorly predictive of clinical outcomesLack of standardized and reproducible protocols and inadequacy to study the early phases of tumor growth (PDX models)Different tumor evolution as compared to parental lesion[90,109,181,186]GeneticallyEngineeredSpecific gene mutationCombination of multiple gene mutationsFunctional immune systemStepwise tumor progressionPhenotypic, histological, and genetic similarities to human counterpartsModulation of human cancer under physiological conditionsTumors develop in the tissue of originInability to replicate the characteristics of the advanced melanomaExpensive, time-consuming and labor intensiveDifferent anatomy, physiology, and biochemistry (mouse versus human)Lack of different genetic background and tissue-specific promotersAsynchronous development of tumors.HeterogeneityRestricted use due to intellectual property rights and patents[90,109,181,186,203,204]Radiation-inducedUseful for studying the risk factors, pathogenesis and development of human melanomaLong time for tumor developmentHigh costs in animal maintenance/careLack of responsiveness by miceHistologically and anatomically different from human melanoma[186]Carcinogen-inducedSimple to accomplishThe tumors are easily visualized, not requiring invasive processes for tumor monitoringRecapitulate the time-dependent and multi-stage progression of tumor pathogenesisFunctional immune systemCan be used in combination with other modelsRepeated use of carcinogenic agentsOutbred mice with non-uniform genetic backgrounds and varied sensitivity to carcinogensNonpigmented lesions when melanoma is induced by certain carcinogenic agentsNot clinically relevant for human melanoma[185,186,203,205]

##### Xenograft Model

Depending on the source material, these models are established in immunocompromised mice by using either human cell lines or tissues obtain from patients (patient-derived xenografts; PDXs). This allows precise and repeated measurements of tumors with the same genetic signature, growing in identical mice and under a strictly controlled environment. In these models, human melanoma cells or tissues are able to interact with the surrounding stroma, namely the circulatory and lymphatic systems, providing valuable insights into human melanoma biology, growth and therapeutic response [75,90]. In xenografts, immunodeficiency is a prerequisite for preventing human tumor cell rejection by the host [75,90,181] and several mouse strains with different immunodeficiency profiles are available [206,207], as summarized in Table 4 and Figure 4b.

Nude (nu/nu) mice were the first to be used in xenograft studies and are characterized by the inability to activate adaptive immune responses. Due to the fact that these animals still possess an intact innate immunity, which hinders primary human tumor engraftment [207], nude mice are most commonly used as cell-based xenografts, since various human cell lines are able to proliferate normally [206]. Additionally, if an extensive protocol is ensued, aged nude mice show “leakage” of small populations of extrathymic T cells that may interfere with tumor establishment, leading to its rejection by the host [206,207]. Regarding melanoma, several recent examples demonstrate the usefulness of nude mice in various research areas, namely melanoma pathophysiology, therapeutic strategies, and prognostic biomarkers (Table 4).

Severe Combined Immunodeficiency (SCID) mice have a mutation in the protein kinase DNA-activated catalytic polypeptide (Prkdc) gene, leading to a defective DNA double-strand break repair during V(D)J recombination and, consequently, absence of mature B and T lymphocytes [206,207,208]. Depending on genetic background, pathogen exposure and age, functional B- and T-cells may accumulate due to a ‘leaky’ phenotype [206]. This condition may, in turn, lead to the development of spontaneous lymphomas, reducing mice life span and limiting their use in prolonged experiments [209]. Although SCID mice have a functional innate immunity and remnant natural killer (NK) cells may limit the success of primary tumor engraftment, this strain is able to host a larger range of primary human tumors compared to nude mice [207,208]. In the context of melanoma research, SCID mice continue to be used as models for studying signaling pathways associated with disease development and progression, as well as for assessing the biodistribution profiles of nanoparticles and therapeutic compounds (Table 4).

To solve the innate immunity problem and improve tumor engraftment, nonobese diabetic (NOD) SCID mice were established. Besides the features of SCID, these animals also display other immune abnormalities, such as loss of complement and defective functions of NK cells, macrophages, and dendritic cells [206,210]. Another strain, the SCID/Beige, has a similar immunological profile to NOD/SCID and, additionally, carries the Lyst^bg^ mutation, which leads to impaired activity of NK cells and macrophages [207,211]. NOD/SCID mice are commonly used for different melanoma studies, as shown by some examples in Table 4.

To further maximize xenotransplantation success, a NOD/SCID strain bearing a mutation in the interleukin-2 receptor gamma chain (NOD/SCID *IL2rg^null^*, NSG) was created [206,212]. These mice completely lack adaptive immunity and present a severe impaired innate immune system, rendering them highly suitable recipients for human stem cells and primary tumor engraftment. In fact, a great majority of human primary tumors have been successfully engrafted into NSG mice, that retain part of the native tumor-infiltrating lymphocytes and stromal cell populations, promoting tumor establishment and growth [206]. In Table 4, several recent examples of melanoma research applications using NSG mice strains are depicted.

All the above-described strains are immunocompromised, limiting their usefulness for therapies involving the human immune system. To overcome this problem, at least to some extent, humanized NSG (hu-NSG) mice have been generated.
cells-10-03088-t004_Table 4Table 4Most frequently used immunocompromised mice for melanoma research.MiceIdentificationMain FeaturesMelanoma Research ApplicationsReferencesNude (nu/nu)AthymicHomozygous for mutation *Foxn1^nu^*T cell deficientHairlessCell line engraftmentPathophysiological mechanismsNovel therapies/therapy resistanceNano-based therapeutic approachesPrognostic biomarkers and molecular imaging[213,214,215,216,217,218,219]SCIDHomozygous for the spontaneous mutation *Prkdc^scid^*T and B cell deficientCell line/tumor engraftmentPathophysiological mechanismsBiodistribution studies[220,221,222,223]NOD/SCIDHomozygous for the spontaneous mutation *Prkdc^scid^*T and B cell deficientImpaired function of macrophages, DC and NK cellsCell line/tumor engraftmentPathophysiological mechanismsGene therapyAdjuvant therapy for brain metastasisDiscovery of novel therapeutic targets[224,225,226,227]NSGNOD/SCID *IL2rg^null^*T and B cell deficientImpaired function of macrophages and DCLack of NK cellsEnhanced tumor engraftmentPathophysiological mechanismsTherapy resistanceNovel therapies/combination therapyChemopreventionDevelopment of imaging probesBiodistribution studiesIdentification of cell subpopulations[217,226,228,229,230,231,232,233,234,235]hu-NSGNSG with humanized immune system induced by CD34^+^ HSC or PBMCPrediction of patients’ response to immunotherapyImaging of therapeutic targetsTherapy resistance[236,237,238,239]DC: dendritic cells; HSC: hematopoietic stem cells; hu-NSG: humanized NSG; IL2rg: interleukin-2 receptor gamma chain; NOD/SCID: nonobese diabetic-severe combined immunodeficiency; NK cells: natural killer cells; NSG: NOD/SCID gamma; PBMC: peripheral blood mononuclear cells; Prkdc: protein kinase DNA-activated catalytic polypeptide.

Cell line xenografts are based on the engraftment of established human melanoma cell lines [75,90]. For tumor induction, cells are most frequently injected s.c. and, to a lesser extent, intradermally. The s.c injection produces a tumor comparable to melanoma skin metastasis. On the other hand, intradermal injection leads to the formation of a tumor that resembles a primary melanoma. However, as mouse skin is thin, the tumor mass may ulcerate, implying early termination of the experimental protocol [90]. Moreover, to establish a metastatic model, human cell lines (e.g., A375 and A2058) can be injected through the i.v. route, usually in the tail vein, and frequently metastasize to the lungs [240,241,242]. Although this procedure allows for rapid spreading of melanoma cells, it does not recapitulate the metastization process that occurs in human disease since, in this case, metastatic cells derive from a primary tumor [90].

To overcome some of the challenges associated with cell line xenografts, PDX models have been established and used for several decades [83], consisting in the implantation of freshly resected human tumor samples into immunocompromised mice [182]. Melanoma PDXs are most frequently generated from metastatic samples of stage III patients because primary tumors are small sized, with all collected samples being used for biopsy or diagnostic purposes [243]. These samples, either from primary or metastatic melanoma, may be directly implanted into the animal (e.g., s.c., intramuscularly) or may be injected as cell suspensions. The latter has demonstrated great efficiency, since the xenotransplantation of few human melanoma cells has led to tumor development [244,245]. In addition, cell suspensions are more representative of the tumor and the PDX generation is less traumatic to mice, contrary to the direct implantation of tumor samples [243].

Compared to cell line-derived tumors, PDXs display a more accurate tumor pathophysiology and genetic heterogeneity, positively impacting and facilitating prognosis and the selection of early clinical-stage patients that would probably respond to new therapies [182]. Moreover, PDX models are useful to define resistance mechanisms of tumors from patients who have relapsed during treatment and help determining alternative therapeutic options [83,246]. In a recent work, researchers established intradermal melanoma PDXs overexpressing an adaptor protein of the Shc family, ShcD. Interestingly, following tumor resection, these models were shown to develop metastasis in different organs, such as lymph nodes, lungs, liver, and spleen. The authors correlated these data with the results from a large cohort of melanoma patients, demonstrating the value of ShcD as prognositic factor and therapeutic target [247].

##### Genetically Engineered Model

Melanoma displays a high mutation burden and is genetically heterogeneous. In this sense, genetically engineered mouse models (GEMMs) (Figure 4c) proved to be very useful for several studies. For example, research involving genetic predisposition, identification of genes involved in melanocyte malignant transformation and melanoma progression and invasion, tumor angiogenesis, evaluation of therapeutic responses and acquired resistance mechanisms [75,248]. Unlike xenograft mice, GEMMs possess a functional immune system and tumors occur spontaneously in the appropriate tumor–stroma tissue or organ microenvironment (Table 3) [90,248]. In addition, these models have been suggested to accurately predict drug efficacy and have also been used for the identification of diagnostic biomarkers [185,249].

GEMMs possess well-validated drug targets, which facilitate the rational design of drug candidates [204]. Indeed, determined genetic aberrations can reproduce the genetic lesions occurring in human melanomas. Common mutations in this context involve, for example, the activation of oncogenes, such as BRAF and RAS and the inactivation of key tumor suppressors, such as PTEN and CDKN2A [90,249]. Thus, Lelliott and collaborators used a syngeneic BRAF V600E Cdkn2a-/-Pten-/- melanoma model and demonstrated that a triple therapy including BRAF, MEK, and CDK4/6 inhibitors led to an immediate tumor regression and improved mice survival, compared to the respective monotherapies [250]. Moreover, BRAF/PTEN-mutated mouse models have also allowed the study of the molecular mechanisms underlying melanoma, namely the role of the Activating Molecule in Beclin-1-Regulated Autophagy in the development of this cancer [251], as well as the identification of new therapeutic targets, such as sirtuin 5 [252]. In addition, novel anticancer vaccines have also emerged as effective therapies towards melanoma as demonstrated in B6-Tyr-Cre^ERT2^BRAF^CA^PTEN^lox/lox^ [253] and B6.Cg-BRAF^tm1Mmcm^ PTEN^tm1Hwu^Tg(Tyr-cre/ERT2)13Bos/BosJ [254] mice.

Furthermore, bitransgenic [255] and hepatocyte growth factor/scatter factor (HGF/SF) transgenic [256] mice have also been used, as well as models including other relevant genes, such as the metastasis suppressor gene NME1 [257], and the oncogene GNAQ [258].

##### Radiation-Induced Model

Models involving the induction of melanomagenesis through UV radiation (Figure 4c) can be useful for simulating the natural initiation and progression of melanoma, since UV exposure is one of the main risk factors for the development of human melanoma [185,259]. However, the different localization of melanocytes between mice and human skin is an important issue to consider, as it will influence the development of melanoma and the response to UV radiation. Indeed, human melanocytes reside primarily in the basal layer of the epidermis and within the epidermal–dermal junction, being susceptible to the penetration by UV radiation. On the other hand, mouse melanocytes are mainly located deeper in the base of hair follicles, being more protected from UV radiation [185,186,260].

One of the most used UV-induced animal models is the HGF/SF transgenic mouse model. In fact, the melanocytes of these mice are located at external sites within the epidermis and at the dermal–epidermal junction, better resembling the human skin. This model has been used for different applications, such as elucidation of molecular mechanisms underlying the melanomagenesis, development of new therapies and prognostic evaluation [256,260]. A GEMM of UV-induced melanoma has also been developed by Pamidimukkala and collaborators. Transgenic mice with C57BL/6 genetic background (HGF, Nme1/2^+/Δ^, Ink4a/p16^−/−^ strains) exposed to an erythematous dose of UV radiation, displayed strong melanin pigmentation, with melanoma mainly located on exposed dorsal skin surfaces, such as the back, neck, top of head, and flanks. In addition, this mouse model also presented lung metastasis and lymph node enlargement, highly correlated across all genotypes [257]. A study conducted in wild-type C57BL/6N mice also showed that chronic exposure to UV radiation induced Melan-A positive cells, a highly selective marker for melanocytes, in the follicular and interfollicular epidermis, as opposed to mice not exposed to UV radiation [261].

The differences in the effects of UVA and UVB on mouse melanomagenesis has been described to be attributed to the mechanisms by which they induce DNA damage. UVB directly damages DNA, leading to the transcriptional activation of melanogenic enzymes. In addition, the development of the tumor has been considered independent of the pigment [185,260,262]. It was found, for example, that UVB induced melanomagenesis through an acute inflammation-dependent process, involving the induction of melanocyte stem cell activation and translocation [263]. On the other hand, UVA causes DNA damage indirectly by stimulating photosensitizers, such as melanin, by generating reactive oxygen species [185,262].

Regarding drug screening studies, the ethyl acetate fraction of *Juniperus communis* inhibited the development of skin pigmentation induced by UVB radiation in HRM-2 melanin-possessing hairless mice. This effect may be associated with a \ reduction of melanin content (37.6%) as a consequence of the reduction of tyrosinase activity (decrease of 48% compared to control) [264]. Moreover, Saba et al. exposed HRM-2 hairless mice to UVB radiation and demonstrated the anti-melanogenic effect of the Korean Red Ginseng Oil, possibly by suppressing the expression of MMP-9 and interleukin-1β. Moreover, the topical application of this natural product decreased the melanin production compared with untreated UVB-induced mice [265]. Other natural products have also demonstrated anti-melanogenic effects. The extract of *Aster spathulifolius* Maxim in UVB-exposed C57BL/6J mice [266] and proanthocyanidins inhibited UV-mediated suppression of immune system, possibly through changes in immunoregulatory cytokines, DNA repair and stimulation of effector T cells [267].

Finally, several mouse models develop tumors after neonatal UV radiation exposure. These are frequently established to understand the influence of age on the onset of malignant melanoma [268,269], assess mutations in critical genes [270], determine skin gene expression (neonatal versus adult mice) and define mechanisms by which melanoma is accelerated by UV radiation [268]. Regarding the later, it was found, for example, that keratinocytic nuclear receptor RXRα provided a protective role, suppressing the formation of spontaneous and acute UVB-induced melanomas, as well as preventing melanoma evolution to more advanced stages in combination with activated CDK4^R24C/R24C^ and oncogenic NRAS^Q61K^ [271].

##### Carcinogen-Induced Model

Beyond the aforementioned models, there are other strategies to initiate melanoma, such as carcinogen-induced tumors (Figure 4c). Several criteria need to be defined for eligibility for skin carcinogenesis regulatory and mechanistic testing, such as a mouse strain that should not develop spontaneous tumors. In addition, both the innate and adaptive immune systems should be intact (unless antitumor immunity is the subject of the study) and the skin should be resistant to irritation to the dose used to induce the tumor [205]. This last criterion is important, since several chemical carcinogens are applied topically, inducing skin irritation and black lesions that further progress into melanoma [185]. An example is 7,12-dimethylbenz(a)anthracene (DMBA), the most potent polycyclic aromatic hydrocarbon derivative. In hairless mice, DMBA produces blue nevus-like spots that are dense dermal deposits of heavily pigmented melanocytes, as proved by histological analysis. These spots have been described as melanocytomas and may develop after a single or multiple applications [272]. Using this approach, Manna et al. skin induced tumors with DMBA in Swiss Albino mice and demonstrated the potential antitumoral effect of a 1,4-dihydropyridine derivative after topical application [273].

A protocol that has been consistently employed is the tumor induction with DMBA followed by chronic promotion with 12-O-tetradecanoyl-phorbol-13-acetate (TPA), acting through the activation of protein kinase C [186]. In this context, a murine model was developed by Nasti and collaborators, using the combination of these two agents. The authors demonstrated the development of melanocytic nevi in C3H/HeN mice and their progression to melanoma. Moreover, they also showed that a single dose of DMBA generated only few and very small nevi and that TPA application alone did not generate nevi. These results highlighted the role of DMBA as an initiator, as well as the importance of the combination of these two agents [274]. Regarding the screening of new drug candidates, several compounds have been evaluated using this combined model (DMBA/TPA) to induce skin cancer, such as the natural compound withaferin A [275] and clinoptilolite-based delivery system loading carmustine [276].

One of the advantages of carcinogen-induced melanoma models (Table 3) is that they are commonly used in combination with other models, such as UV radiation, GEMMs and xenotransplantation in order to better resemble the human condition [185,186].

#### 3.3.2. Zebrafish Model

Although mice are undoubtedly the most frequently used species in cancer research, other animal models are available. In this context, zebrafish (*Danio rerio*) has emerged as a useful in vivo model due to the natural transparency of the embryos, and the ability to follow the fate of fluorescent-labeled cancer cells using high-resolution imaging. Other advantages of this model include the short generation time, the large number of progeny, the deficient immune system that enables easy xenotransplantation of human cancer cells, the ease of genetic manipulation, their low cost, small housing, and suitability for HTS [90,277,278,279]. In addition, the cancer cell extravasation process in zebrafish is similar to humans, since cancer cells are frequently trapped in the capillaries, and interaction with the surrounding endothelium and the active remodeling of endothelial structures are required for their extravasation [278].

Zebrafish models have been used for different purposes, such as to model melanoma pathology drivers, to investigate genetic modifiers of the disease and drug resistance mechanisms, and to understand the melanocyte development and melanoma microenvironment, initiation and progression [280,281,282,283]. Regarding drug screening experiments, several studies using zebrafish can be found in the literature, demonstrating the usefulness of this model. An example is the study conducted by Mahmood et al., who showed that either mutated or wild type Shiitake extracts exerted anti-melanin activity. This effect resulted from the reduction of pigmentation, as observed on the tail of the zebrafish larvae, stopping the embryogenesis process only at concentrations higher than 900 μg/mL [284]. In addition, in zebrafish embryos, gedunin, a naturally occurring heat-shock protein 90 inhibitor, also demonstrated its anti-melanogenesis activity, reducing the number of pigment dots and melanin content, without eliciting toxicity or morphological abnormalities at concentrations ranging from 25–100 µM [285]. Other tested compounds using zebrafish models were recently demonstrated to possess antimelanoma potential. Among them are diphlorethohydroxycarmalol [286], theaflavin [287], ellagic acid [288], inularin [289], and BEL β-trefoil [290].

Despite the utility and advantages of zebrafish, this model also presents some limitations. Its physiology is different from humans, compounds are administered via distinct routes (e.g., dissolution of drug candidates in egg water), the existence of the chorion may interfere with drug diffusion, and there are technical challenges associated with cancer cell injection and live imaging of intact organisms [46,278].

#### 3.3.3. Canine Models

Melanoma is a common disease in dogs, affecting different anatomical sites, including the lips, oral/mucosal cavity, skin, eyes, and footpad/nails. Among these, oral melanoma is the most aggressive type [291,292,293]. This pathology has been associated with breed predisposition and has shown a higher incidence in cocker spaniels, poodles, pekinese, Gordon setters, chow-chows, golden retrievers, Scottish terriers, dachshunds and mixed breed dogs [109].

Canine models present several advantages over murine ones in oncology field. Dogs exhibit spontaneous, highly aggressive tumors with the same anatomic and physiologic characteristics of human tumors, developing over long periods of time in the presence of a functional immune system. In addition, the inter-individual and intra-tumoral heterogeneity, development of drug resistance, recurrences and metastasis also closely mimic the progression of human tumors. Moreover, dogs live in the same environment and are exposed to the same carcinogens as humans [109,294]. Regarding cutaneous melanoma, this type of cancer has demonstrated a relatively benign behavior in dogs and, contrary to humans, UV light exposure does not play a significant role in the canine disease (since they have a protective hair coat), suggesting a different pathophysiological mechanism [109,294,295]. Additionally, canine cutaneous melanoma develops more commonly in black-coated breeds with pigmented skin, as opposed to what is observed in human cutaneous melanoma, where individuals with light skin are at higher risk [296].

#### 3.3.4. Other Animal Models

In addition to the aforementioned in vivo models, there is a panoply of other animal species that can be used to study melanoma. A relevant model is the chicken chorioallantoic membrane that consists of a highly vascularized extra-embryonic membrane [278,297]. The good visibility, accessibility, and cost-effectiveness of this model turns it a suitable candidate for evaluating cancer biology as well as performing HTS in vivo. The i.v. injection of fluorescence-labeled cancer cells enables researchers to easily distinguish between the intravascular cancer cells and the extravasated counterparts using confocal microscopy. Furthermore, as described in literature, the injection of melanoma cells reproduces the invasive features of the human disease [277,278,298,299]. Some drawbacks associated with this model are the continuous and rapid morphological alterations that occur even during short experimental periods and the inability to analyze the multi-step processes of metastasis. Importantly, compared to mammalian models, avian species may limit the number of reagents compatible with the model, including antibodies and cytokines [277,278]. Swine is another relevant animal model of melanoma, due to the considerable resemblance to humans in terms of genetics, physiology and anatomy. For example, opposed to mice, the location of melanocytes and the lifespan are similar to humans, allowing the study of long-term efficacy and toxicity of anticancer agents [46,300,301,302]. Hamsters (e.g., Syrian hamsters) have also been used for several purposes, such as preclinical efficacy studies, biodistribution and toxicity of new therapies, as well as their subjacent antimelanoma mechanisms [303,304]. An interesting advantage of this particular model is the variability of hamsters hair-coat coloration phenotypes. This enables the studying of genetics and the influence of hair color phenotypes on melanoma development. However, these animals are resistant to UV radiation-induced melanomagenesis [305]. Other melanoma animal models include the gerbil [305], horse [306], opossum [307] and other fish species, such as Xiphophorus [308] and medaka [309].

## 4. Ongoing Clinical Trials

Aiming to compile ongoing melanoma clinical trials, the database ClinicalTrials.gov (https://clinicaltrials.gov (accessed on 13 June 2021)) was used as source of information. The search was performed for each melanoma stage (from I to IV). The study status (Recruiting, Not yet recruiting, and Active not recruiting) as well as type (Interventional) were selected to refine the search. Although for each melanoma stage several clinical trials were listed, not all of them were selected as they did not fit the defined criteria. The results of this search are summarized in Figure 5.

For melanoma stages I, II, III and IV, 5, 21, 143 and 288 clinical trials were retrieved, respectively. As observed in Figure 5, the high number of clinical trials found for melanoma stages III and IV may be explained by the reduced effective therapies in clinical use thus prompting the investigation for better therapeutic options of such complex and aggressive skin cancer.

For melanoma stage II, among the 21 studies found, the majority of the clinical trials were in phase 3 (33%) and 2 (29%), respectively, and 20% started in 2019. Regarding melanoma stage III, 143 clinical trials were eligible in the present analysis. Here, the majority of clinical trials were in phase 2 (45%) or phase 1/2 (13%) that include safety assessment and therapeutic efficacy evaluation. Clinical trials phase 1 represented 27% of the total studies.

For melanoma stage IV the highest number of clinical trials, totaling 288 were retrieved. At this advanced melanoma stage, again most of clinical trials are in phase 2 (39%), or phase 1/2 (14%) [310].

Some examples of ongoing clinical trials for each melanoma stage are listed in Table 5.

To summarize, our analysis included a total of 457 ongoing clinical trials, of which 63% corresponded to melanoma at stage IV, 31% to melanoma stage III, and 5 and 1% to stages II and I, respectively. The high number of clinical trials for melanoma stages III and IV reflects the investigational efforts to improve the poor clinical outcomes of the severity, aggressiveness and treatment resistance of the disease.

Regarding the clinical trial phase (Figure 6), the majority are in phase II (40%), followed by phase I (30%), corresponding to the evaluation of drug effectiveness and safety assessment, respectively [310].

## 5. Approval for Marketing by Regulatory Agencies

Treatments for melanoma have advanced on two fronts: immunotherapy and targeted therapy. Nevertheless, the majority of patients with metastatic melanoma do not respond to treatment or are prone to secondary resistance, with later disease progression. Therefore, the combination of these therapeutic modalities with chemotherapy has been explored as an advantageous strategy for melanoma management [311].

The first clinical trial on melanoma started in 1971 in Argentina, followed by the FDA approval of dacarbazine in 1975. Later on, several other drugs were approved by the FDA for the treatment of melanoma, such as IL-2, ipilimumab, peginterferon alfa-IIb, dabrafenib, trametinib, pembrolizumab, nivolumab, talimogene laherparepvec and vemurafenib. Besides these isolated drugs, other drug combinations like trametinib with dabrafenib or nivolumab with ipilimumab or vemurafenib with cobimetinib or encorafenib with binimetinib [312,313] have been also approved by FDA. More recently, the EMA has validated the marketing authorization application for the fixed-dose combination of relatlimab and nivolumab in the frontline treatment of adult and pediatric patients with advanced melanoma.

The full research, drug development and approval process of a new medicine is a long process. Fortunately, the regulatory agencies, like the FDA, offer an Accelerated Approval Program and a Fast-Track Program to speed up the approval process. In fact, successful drug approval requires adequate and well-controlled studies demonstrating that the drug is effective and safe for its intended use. In this sense, a pharmaceutical company seeking regulatory authorization must complete a five-step process, as depicted in Figure 7: Early drug discovery, Preclinical research, Clinical development, Clinical approval and Post-market surveillance. First, laboratory tests and preclinical studies must be performed using various animal models to evaluate the biological activity and safety of the drug candidate. In addition, the pharmaceutical company must ensure proper manufacturing conditions of the drug candidate, including composition, stability, and batch-to-batch reproducibility. Then, based on the success of the obtained data, the investigational new drug is enrolled in clinical trials, namely: Phase 1, where the main goal is safety assessment; Phase 2, where drug effectiveness is confirmed; and Phase 3, which aims to achieve deeper information on the safety and effectiveness of the drug in a larger number of patients.

After these steps, regulatory officials will review the information and, if the findings show a positive balance between benefit and risk, the drug is clinically approved. Nevertheless, even after successfully entering the market, regulatory agencies will continue to monitor the drug post-approval (post-market surveillance).

However, in the cases where regulatory entities refuse the approval, the pharmaceutical enterprise will have the opportunity to discuss and to correct the problems with regulatory officials and, if applicable, submit new information or, in the worst scenario, withdraw the application. Common issues that may lead to rejection include administrative problems (e.g., lacking important data), unpredicted safety problems, or failure to show drug effectiveness. Regarding safety and efficacy issues, the pharmaceutical company may need to conduct studies in a larger and more diverse populations. Another option can include studies that span longer periods of time. Since the manufacturing process may also lead to delay or rejection of market approval, the manufacturing facilities are inspected in order to ensure that drug production follows the required standards, Good Manufacturing Practices.

For melanoma research, it is obvious that tremendous scientific progress in drug approval has been made. More efforts have also been made to understand how to explore the complexity of this disease and, hopefully, translating it into unprecedented clinical success. Despite the major hurdles, the challenges are now clearly defined and are being actively investigated. We know that treatment outcomes are now more promising than ever, but more research still needs to be done.

## 6. Conclusions

Several preclinical models have been described aiming to investigate melanoma and design more effective and safe therapies targeting this pathology. Drug discovery and development programs usually adhere to the following steps: (i) initial use of in silico techniques in order to predict and identify antimelanoma candidates that show the best probability of success regarding their physicochemical, pharmacodynamic, and pharmacokinetic properties; (ii) HTS in 2D cultures using a variety of cancer cell lines or primary cells; (iii) from the 2D experiments, the most promising candidates are selected and move forward to more sophisticated in vitro studies, 3D cultures; (iv) considering all these data, the last steps of preclinical studies involve the use of adequate animal models for safety assessment and therapeutic evaluation; (v) finally, drug candidates proceed to the more cost-, time- and labor-intensive clinical trials and, eventually, may successfully reach the market.

Since cancer was initially identified as a disease of uncontrolled cell growth and division, research attempts have been mainly directed to the discovery of antiproliferative agents, with cytotoxicity/antiproliferative assays being the most used in the context of anticancer discovery. This has led researchers to perform in vitro screening using different cancer cell lines, with distinct characteristics. Currently, in vitro assays that recreate tumor progression, heterogeneity, plasticity, migration, invasion, and interactions with the microenvironment are available. These important features are more evident in 3D models, which better resemble the human melanoma and reduce the limitations of the basic and simpler in vitro 2D cultures. Undoubtedly, in vitro models provide important data about drug biological activity, target interaction, and mechanistic action. Moreover, in vitro screening allowed the reduction of the number of animals used in the next steps of drug development, respecting the 3R’s rule. However, despite the recent advances, these studies still do not entirely mimic melanoma’s complexity, organization, and microenvironment in vivo. For this reason, animal models are a key piece of the drug development program.

Indeed, mice are suitable models to elucidate the relevance aspects of melanomagenesis and are an indispensable tool for the preclinical testing of drug efficacy and safety, as well as resistance mechanisms. Although robust animal models have been developed over the years, such as PDXs or GEEMs, the majority of the experimental protocols rely on syngeneic mice models or xenografts induced with commercial human melanoma cell lines. Despite their utility and considering the complexity of melanoma, preclinical mouse models have demonstrated poor prognostic value and have shown high rates of failure in their translation to human clinics. Nonetheless, the fact is that a considerable number of antimelanoma therapies have reached clinical trials. In this context, most of the studies are directed to advanced stages of melanoma, since it is associated with higher aggressiveness, therapy resistance and mortality rates. On the other hand, the first stages of melanoma are frequently treated with surgical procedures, which can justify the lower number of studies found for melanoma stages I and II.

In the context of innovative therapeutic strategies, chimeric antigen receptor (CAR)-T cell therapy has demonstrated improved clinical outcomes, in comparison with standard chemotherapy or immunotherapy. CAR-T cells lead to tumor cells apoptosis and lysis after the release of several cytotoxic molecules and cytokines due to the recognition of target antigens located on the surface of cancer cells, independent of major histocompatibility complex participation. Although this type of therapy has revealed interesting results in preclinical and even clinical trials, efforts should be made to improve its safety profile and reduce the costs of production [314]. The oncolytic virotherapy is another recent therapeutic strategy for the management of melanoma. This therapy is constituted by a genetically engineered virus, which selectively replicates in tumor cells, causing their lysis and promoting antitumor immunity. Oncolytic viral immunotherapy (T-VEC) was already approved for the local treatment of unresectable metastatic melanoma at stages III and IV [315]. This opened new doors for the development of new vaccines including these viruses.

Another hot topic is the necessity to search for new melanoma biomarkers that effectively help the diagnosis, prognosis, and prediction of treatment responses. Serum biomarkers, such as lactate dehydrogenase and S100 β are well established regarding the prognosis and monitoring of melanoma, although their role at stage III resected or metastatic melanoma is not clear [316,317]. Moreover, melanoma-inhibiting activity and vascular endothelial growth factor have been associated with advanced stages of the malignancy, but have presented low specificity. Genetic biomarkers, such as BRAF and NRA, have also afforded associations regarding selecting patients and predicting responses to target therapy. The establishment of new and effective biomarkers remains of high importance, particularly in the time of personalized medicine [316].

Globally, several resources are being invested in the prevention, diagnosis, and treatment of melanoma. Some of the key challenges involved in the anticancer drug discovery include difficulty in identifying promising therapeutic targets, selection of appropriate preclinical screening models, and high financial costs. The decision on which technique should be used during early drug discovery and development process may be guided by the advantages and limitations that were described in this review. Indeed, the ideal approach should consider the use of different and complementary models to obtain more robust and reliable information before the compounds enter clinical trials.

## Figures and Tables

**Figure 1 cells-10-03088-f001:**
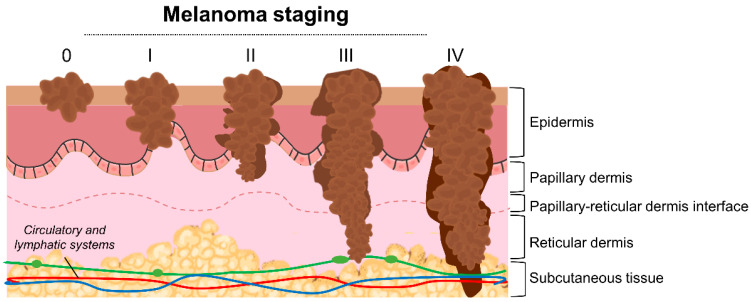
Melanoma staging, according to the TNM system defined by the AJCC. Part of the figure was drawn using pictures from Servier Medical Art. Servier Medical Art by Servier is licensed under a Creative Commons Attribution 3.0 Unported License (https://creativecommons.org/licenses/by/3.0/ (accessed on 20 June 2021)).

**Figure 2 cells-10-03088-f002:**
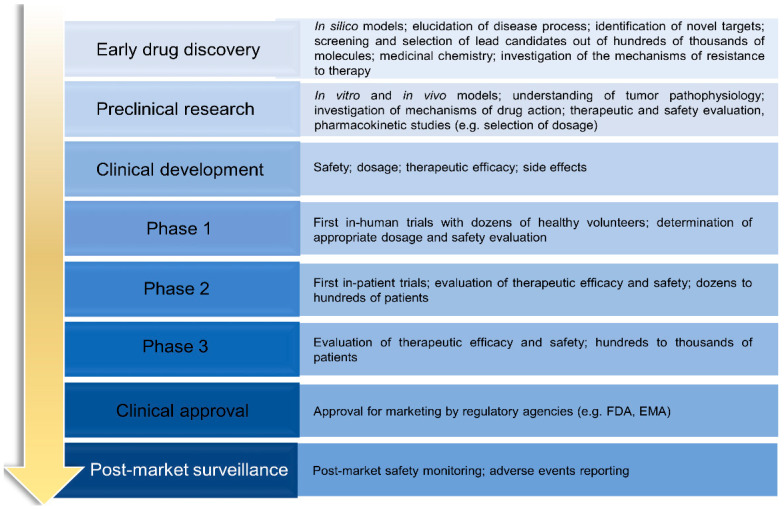
Drug development steps, from early drug discovery to post-market surveillance.

**Figure 3 cells-10-03088-f003:**
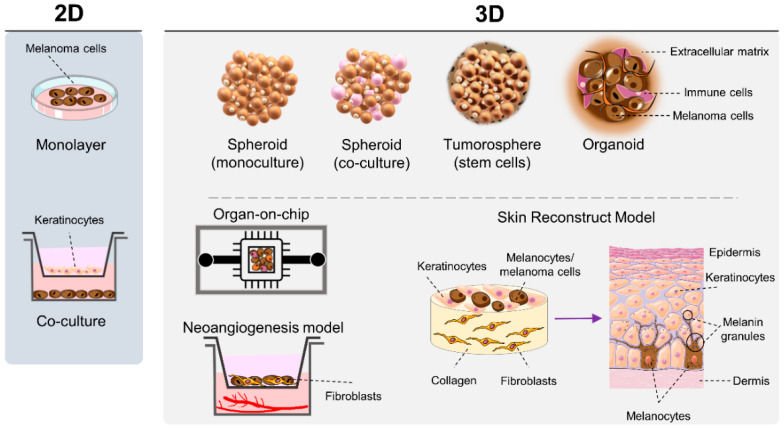
Schematic representation of the main in vitro models used for melanoma research. Part of the figure was drawn by using pictures from Servier Medical Art. Servier Medical Art by Servier is licensed under a Creative Commons Attribution 3.0 Unported License (https://creativecommons.org/licenses/by/3.0/ (accessed on 25 June 2021)).

**Figure 4 cells-10-03088-f004:**
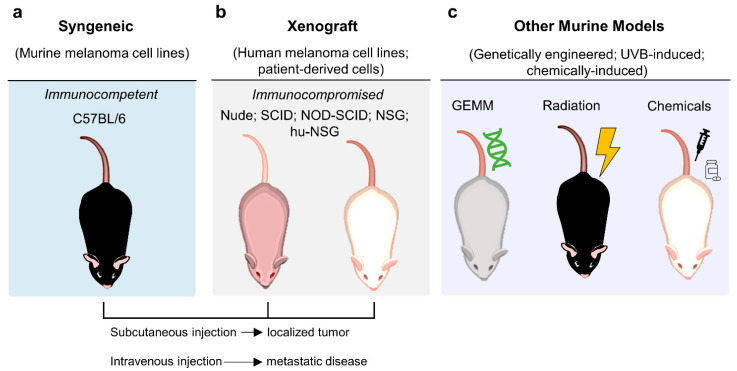
Main murine models used for melanoma research: (**a**) syngeneic, (**b**) xenograft and (**c**) genetically engineered, UVB-induced and chemical-induced.

**Figure 5 cells-10-03088-f005:**
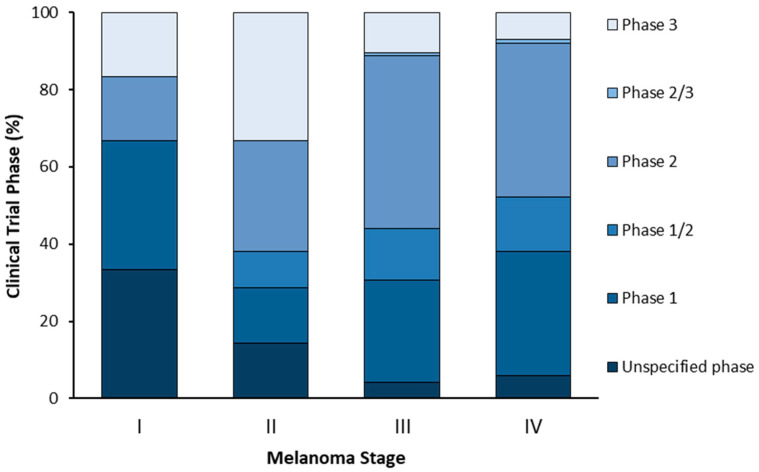
Distribution of melanoma clinical trials phases, in percentage, according to disease stage. Data obtained from https://clinicaltrials.gov, accessed on 13 June 2021.

**Figure 6 cells-10-03088-f006:**
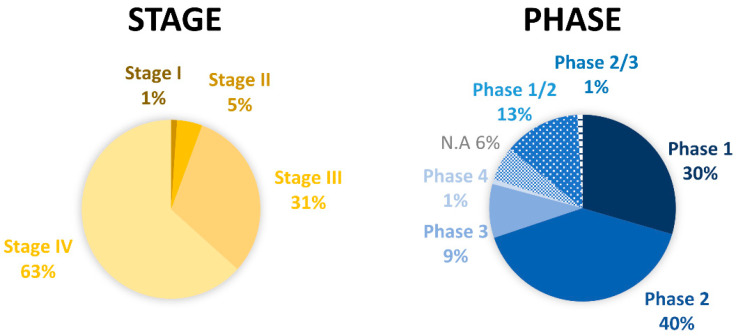
Graphical representation of clinical trials by melanoma disease stage and by clinical phase. Data obtained from https://clinicaltrials.gov, accessed on 13 June 2021.

**Figure 7 cells-10-03088-f007:**
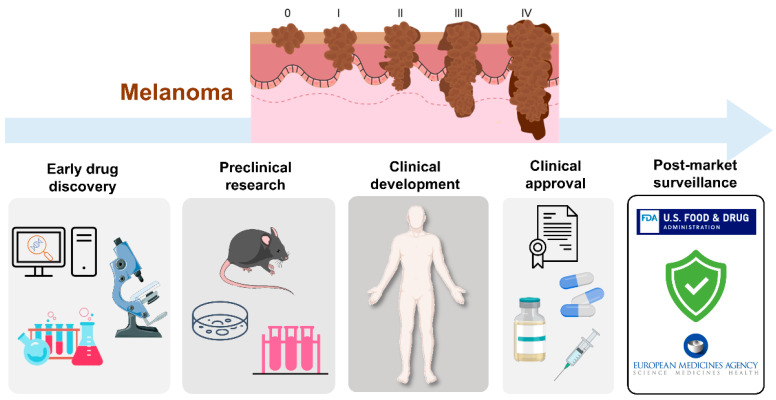
Schematic representation of melanoma drug development pipeline.

**Table 1 cells-10-03088-t001:** Main advantages and limitations of the currently available therapeutic options for melanoma.

Therapeutic Option	Advantages	Limitations
Surgery	Prevention of melanoma systemic spreading Reduction of local recurrence risk	Associated severe comorbidities Unsuitable for systemic disease
Radiotherapy	Good local tumor control Useful for palliative care	Associated intrinsic resistance Severe side effects
Chemotherapy	Effective in highly proliferating disease conditions Indicated for palliative care	Reduced specificity Severe side effects
Targeted therapy	Reduction of side effects Improvement of response and survival rates Personalized therapy	Emergence of resistances High cost
Immunotherapy	Improvement of clinical outcomes	Severe side effects High cost

**Table 2 cells-10-03088-t002:** Main in vitro melanoma models: advantages and disadvantages.

Experimental Model	Advantages	Disadvantages	References
2D models	Initial screening and selection of new molecules Basic research of tumor cell biology Easy to perform Compliant with HTS High reproducibility Cost-effective Pure and free from contaminating cells	Unable to mimic in vivo tumor microenvironment Loss of stromal, vascular, and immune cellular populations Lack of heterogeneity Alterations after long-term culture Cannot reproduce melanoma cell interactions with extracellular matrix	[90,108,109]
3D models			
Spheroid	Easy to perform Compliant with HTS Co-culture ability Relative low cost High reproducibility Good representation of oxygen, nutrient, and other soluble factors Simulation of tumor heterogeneity and drug resistance	Simplified architecture Unsuitable for longitudinal studies Limited number of cell types for co-culture	[75,110,111,112]
Tumorosphere	Preservation of cancer stem cell features Initiating ability Self-renewal potential Study of drug resistance	Excessive sensitivity to the culture method Cell fusion and aggregation Low reproducibility Unable to reproduce the variety of cell types	[74,111,113,114]
Organoid	Compliant with HTS Patient specific Simulation of in vivo tumor complexity and architecture	Low reproducibility Lack of vascular system and/or key cell types Reduced number of cells available Reduced heterogeneity compared to original tumor	[75,110]
Skin Reconstruct	Simulation of in vivo tumor architecture Controlled tissue organization Co-culture ability	Time-consuming procedure Constant monitoring required	[111,115]
Melanoma-on-chip	Simulation of in vivo tumor architecture, microenvironment, chemical and physical gradients Accurate and rapid procedure Cost-effective (small scale)	Lack of vascular system Unsuitable for HTS	[74,110]
Neoangiogenesis	Simulation of in vivo tumor microenvironment Co-culture ability Time-effective	Insufficient vessel stabilization due to the short duration of assays Formation of vessel like-structures instead of capillaries	[75,116,117]

**Table 5 cells-10-03088-t005:** Examples of ongoing clinical trials for each melanoma stage.

Clinical Trial (NCT Number)	Melanoma Stage	Clinical Phase	Start Date	Sponsor
NCT04697576	I/II/IV	1	2021	Carlo Contreras
NCT03819296	I/II/III/IV	1/2	2021	M. D. Anderson Cancer Center
NCT03757689	II	2	2019	Abramson Cancer Center
NCT03860883	II	3	2019	Melanoma and Skin Cancer Trials Ltd.
NCT04309409	II	3	2020	University Hospital, Essen
NCT03554083	III	2	2018	Mayo Clinic
NCT03021460	III/IV	1	2017	Mayo Clinic
NCT03132675	III/IV	2	2017	OncoSec Medical Inc.
NCT02816021	III/IV	2	2017	M. D. Anderson Cancer Center
NCT03991130	III/IV	2	2019	Gregory Daniels
NCT04356729	III/IV	2	2020	Elizabeth Buchbinder
NCT02506153	III/IV	3	2015	National Cancer Institute
NCT04410445	III/IV	3	2020	Nektar Therapeutics
NCT01993719	IV	2	2013	National Cancer Institute
NCT03928275	IV	2/3	2021	Carman Giacomantonio

Data obtained from https://clinicaltrials.gov, accessed on 13 June 2021.

## Data Availability

Data presented in this study are available on request from the corresponding author.

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
