# Peer review of "The Challenging Melanoma Landscape: From Early Drug Discovery to Clinical Approval"

_cells, 2021, doi:10.3390/cells10113088_

Round 1

Reviewer 1 Report

This manuscript is a sharp and comprehensive review of the recent literature regarding all the steps involved in the process of antimelanoma agents develop and selection. I found it very clear and straightforward, especially the sections regarding the preclinical research state of the art, including the description of so many different in vitro and in vivo melanoma models. In general, it gives a complete view of the current scenario in the field. I found it pleasant to read and also very useful for scientist working on all the different faces of melanoma research, in both basic and applied science fields. Figures, diagrams and Tables are also very explicative and clear.

My compliments to the authors

To my opinion is very well written and I absolutely recommend to publish it in the present form.

Author Response

Reviewer 1

This manuscript is a sharp and comprehensive review of the recent literature regarding all the steps involved in the process of antimelanoma agents develop and selection. I found it very clear and straightforward, especially the sections regarding the preclinical research state of the art, including the description of so many different in vitro and in vivo melanoma models. In general, it gives a complete view of the current scenario in the field. I found it pleasant to read and also very useful for scientist working on all the different faces of melanoma research, in both basic and applied science fields. Figures, diagrams and Tables are also very explicative and clear.

My compliments to the authors

To my opinion is very well written and I absolutely recommend to publish it in the present form.

REPLY: we are very grateful for the comments  of reviewer 1.

Reviewer 2 Report

The authors need to define what audience they are targeting. It reads like an introduction to a Ph D thesis.  The article is too long and needs to be more concise throughout. They could drop out the first section on treatment of melanoma and go straight to the various preclinical  approaches. The article would be useful if it just focused on preclinical approaches.  Clinical trials is a big subject and rather pointless trying to cover this.

Author Response

Reviewer 2

The authors need to define what audience they are targeting. It reads like an introduction to a PhD thesis.  The article is too long and needs to be more concise throughout. They could drop out the first section on treatment of melanoma and go straight to the various preclinical approaches. The article would be useful if it just focused on preclinical approaches.  Clinical trials are a big subject and rather pointless trying to cover this.

REPLY: We appreciate this valuable comment. In this regard, Section 2 “Melanoma – Therapeutic Management” has been substantially reduced, retaining only major aspects related with melanoma therapy. Section 4 “Ongoing Clinical Trials” was included in the article to provide a general idea of current progress and success of drug development for melanoma treatment.

Reviewer 3 Report

The review entitled “The challenging melanoma landscape: from early drug discovery to clinical approval” by Matias et al addresses key aspects regarding the current and future therapeutic approaches in melanoma. This is a quickly growing field that needs to be reviewed timely. In my opinion, a few aspects of the manuscript need more attention by the Authors, namely:

  • 1) Line 57-71: Authors report the Clark level staging system, with no mention of the Breslow depth and no mention of the currently approved staging systems. The current staging system approved by the American Joint Committee on Cancer (AJCC) does not consider Clark Level as a good staging system., currently. I suggest Authors should clearly refer to the currently approved AJCC staging of melanoma. For completeness, along with Clark level, Authors may also cite the staging system based on Breslow depth.
  • 2) Figure 1: Please substitute this figure with a Figure referring to the current AJCC staging system
  • 3) Line 85-91: Some details on the PubMed screening performed are lacking. For instance, the actual date by which the PubMed search has been performed. This information is useful to the reader, to be advised on the period to which the search is referred to. Also, the actual fields of the searches are lacking (“All fields”, or Title, or “Abstract”, or “MESH terms”, etc).
  • 4) I believe the role of diet and dietary alterations should be mentioned in this review. In PubMed, several studies are present, regarding the role of the diet in the melanoma management as well as the role of vitamin D and other vitamins. The role of the diet and of specific dietary constituents as risk factors or protective factors or supportive therapeutic approaches, deserve to be at least mentioned, in my opinion.
  • 5) Car-T approach is not mentioned. Many hopes are currently raising from this approach. Several studies have been published on CAR-T and Melanoma, that deserve to be mentioned, in my opinion, and discussed in an ad-hoc paragraph.
  • 5) The large efforts on the identification of novel effective biomarkers potential therapeutic targets merit to be at least mentioned and discussed in an ad-hoc paragraph, as well as recent advances on "oncolytic virus" approaches

Author Response

Reviewer 3

1) Line 57-71: Authors report the Clark level staging system, with no mention of the Breslow depth and no mention of the currently approved staging systems. The current staging system approved by the American Joint Committee on Cancer (AJCC) does not consider Clark Level as a good staging system., currently. I suggest Authors should clearly refer to the currently approved AJCC staging of melanoma. For completeness, along with Clark level, Authors may also cite the staging system based on Breslow depth.

REPLY: Thank you for your comment. The information regarding melanoma staging has been corrected accordingly.

“Classically, melanoma progression has been represented by the Clark model, a previously widely accepted method for melanoma microstaging, based on the anatomic level of local invasion [20-22]. Nowadays, the staging of melanoma considers several parameters, including the Breslow depth, ulceration, extent of regional lymph nodes invasion, and degree of spreading in the surrounding area and distant parts of the body. As represented in Figure 1, the American Joint Committee on Cancer (AJCC) has defined the TNM system – tumor, lymph nodes, metastasis – for melanoma staging, being internationally recognized and commonly applied. Briefly, in the TNM system: (i) T refers to the size of the primary tumor and its spreading to adjacent tissues; (ii) N describes the number of regional lymph nodes affected by the tumor; and (iii) M identifies the presence of metastasis. At stage 0, abnormal melanocytes are observed in the epidermis, being designated as melanoma in situ. At stages I and II, localized cancer has formed. At stage III, cancer has spread to nearby lymph nodes, and at stage IV melanoma has spread to other tissues/organs, namely lung, liver, brain, spinal cord, bone, soft tissue, gastrointestinal tract, and distant lymph nodes.”

2) Figure 1: Please substitute this figure with a Figure referring to the current AJCC staging system.

REPLY: Figure 1 was replaced with one representing the AJCC staging system.

3) Line 85-91: Some details on the PubMed screening performed are lacking. For instance, the actual date by which the PubMed search has been performed. This information is useful to the reader, to be advised on the period to which the search is referred to. Also, the actual fields of the searches are lacking (“All fields”, or Title, or “Abstract”, or “MESH terms”, etc).

REPLY: Thank you for the suggestions. We have included in the article more details concerning the details that were used for writing the present review. 

“To prepare this review, an extensive literature search was performed using electronic resources. PubMed and Science Direct were the main sources of information, complemented by other information sources, such as Research Gate, and the use of official sources from European Medicines Agency (EMA) and Food and Drug Administration (FDA). The search was carried out between January 2021 and June 2021, aiming to generate a critical and comprehensive overview of the methodologies used for antimelanoma drug discovery and development, as well as for the study of the mechanisms underlying this pathology. Research was occasionally carried out outside these dates. From the articles collected from the initial literature search, an analysis was carried out to select the most relevant ones. The keywords for the search under “Title/Abstract”consisted of combinations of the following terms: melanoma, drug development, computational studies, in vitro, 3D assays, in vivo and animal models. The search regarding clinical trials is described in the respective section.”

4) I believe the role of diet and dietary alterations should be mentioned in this review. In PubMed, several studies are present, regarding the role of the diet in the melanoma management as well as the role of vitamin D and other vitamins. The role of the diet and of specific dietary constituents as risk factors or protective factors or supportive therapeutic approaches, deserve to be at least mentioned, in my opinion.

REPLY: Thank you for your comment. Information regarding the role of diet in melanoma management was included in the Introduction section.

“Moreover, the dietary regimens have also attracted attention for reducing melanoma risk. Several antioxidant phytochemicals from vegetables and fruits, for example, have demonstrated chemopreventive properties, whereas alcohol intake can increase the risk of malignancy [9,10]. The role of vitamin D in the management of melanoma should also be highlighted, due to its antiproliferative effects. In this sense and since solar radiation is essential for vitamin D production, a careful sun exposure is recommended [9].”

5) Car-T approach is not mentioned. Many hopes are currently raising from this approach. Several studies have been published on CAR-T and Melanoma, that deserve to be mentioned, in my opinion, and discussed in an ad-hoc paragraph.

REPLY: The information about CAR-T cell therapy was included in the conclusion section.

“In the context of innovative therapeutic strategies, chimeric antigen receptor (CAR)-T cell therapy has demonstrated improved clinical outcomes, in comparison with standard chemotherapy or immunotherapy. CAR-T cells lead to tumor cells apoptosis and lysis after the release of several cytotoxic molecules and cytokines due to the recognition of target antigens located on the surface of cancer cells, independently of the major histocompatibility complex participation. Although this type of therapy has revealed interesting results in preclinical and even clinical trials, efforts should be made to improve its safety profile and reduce the costs of production [316].”

6) The large efforts on the identification of novel effective biomarkers potential therapeutic targets merit to be at least mentioned and discussed in an ad-hoc paragraph, as well as recent advances on "oncolytic virus" approaches.

REPLY: Additional information regarding biomarkers and oncolytic virotherapy was included in the conclusions section.

“The oncolytic virotherapy is another recent therapeutic strategy for the management of melanoma. This therapy is constituted by a genetically engineered virus, which selectively replicates in tumor cells, causing their lysis and promoting antitumor im-munity. Oncolytic viral immunotherapy (T-VEC) was already approved for the local treatment of unresectable metastatic melanoma at stages III and IV [317]. This opened new doors for the development of new vaccines including these viruses.

Another hot topic is the necessity to search new melanoma biomarkers that effectively help the diagnosis, prognosis and prediction of treatment responses. Serum biomarkers, such as lactate dehydrogenase and S100 β are well established regarding the prognosis and monitoring of melanoma, although their role at stage III resected or metastatic melanoma is not clear [318,319]. Moreover, melanoma-inhibiting activity and vascular endothelial growth factor, have been associated with advanced stages of the malignancy, but have presented low specificity. Genetic biomarkers, such as BRAF and NRA, have also afforded associations regarding selecting patients and predicting responses to target therapy. The establishment of new and effective biomarkers remains of high importance, particularly in the times of personalized medicines [318].”